# Perceived needs of disease vector control programs: A review and synthesis of (sub) national assessments from South Asia and the Middle East

Henk van den Berg[1], Kabirul Bashar[2], Rajib Chowdhury[3,4], Rajendra M. Bhatt[5], Hardev Prasad Gupta[6], Ashwani Kumar[7,8], Shanmugavelu Sabesan[7], Ananganallur N. Shriram[7], Hari Kishan Raju Konuganti[7], Akhouri T. S. Sinha[9], Mohammad Mehdi Sedaghat[10], Ahmadali Enayati[11], Hameeda Mohammed Hassan[12], Aishath Shaheen Najmee[13], Sana Saleem[13], Surendra Uranw[14], Pahalagedera H. D. Kusumawathie[15], Devika Perera[15], Mohammed A. Esmail[16], Lauren B. Carrington[17], Samira M. Al-Eryani[18], Roop Kumari[19], Bhupender N. Nagpal[20], Sabera Sultana[4], Raman Velayudhan[21], Rajpal S. Yadav[21,22]*

1 Laboratory of Entomology, Wageningen University, Wageningen, the Netherlands, 2 Department of Zoology, Jahangirnagar University, Savar, Dhaka, Bangladesh, 3 Nutrition and Clinical Services Division, International Centre for Diarrhoeal Disease Research, and Department of Public Health, Independent University Bangladesh, Dhaka, Bangladesh, 4 World Health Organization Country Office for Bangladesh, Dhaka, Bangladesh, 5 Consultant, Nadiad, Gujarat State, India, 6 Consultant, Guwahati, Assam, India, 7 ICMR-Vector Control Research Centre, Puducherry, India, 8 Saveetha Institute of Medical and Technical Sciences, Saveetha University, Chennai, India, 9 Consultant, Ranchi, Jharkhand, India, 10 School of Public Health, Tehran University of Medical Sciences, Tehran, Iran, 11 School of Public Health, Mazandaran University of Medical Sciences, Sari, Iran, 12 Vector Control Section, Department of Public Health, Ministry of Health, Baghdad, Iraq, 13 Consultant, Male, Maldives, 14 B. P. Koirala Institute of Health Sciences, Dharan, Nepal, 15 Retired Regional Malaria Officer, Colombo, Sri Lanka, 16 National Malaria Control Program, Ministry of Public Health & Population, Sana'a, Yemen, 17 Global Malaria Programme, World Health Organization, Geneva, Switzerland, 18 Regional Office for the Eastern Mediterranean, World Health Organization, Cairo, Egypt, 19 World Health Organization Country Office for India, New Delhi, India, 20 Regional Office for South-East Asia, World Health Organization, New Delhi, India, 21 Veterinary Public Health, Vector Control and Environment Unit, Department of Control of Neglected Tropical Diseases, World Health Organization, Geneva, Switzerland, 22 Academy of Public Health Entomology, Udaipur, India

* rajpal@yadav.cloud

## Abstract

Systems for disease vector control should be effective, efficient, and flexible to be able to tackle contemporary challenges and threats in the control and elimination of vector-borne diseases. As a priority activity towards the strengthening of vector control systems, it has been advocated that countries conduct a vector-control needs assessment. A review was carried out of the perceived needs for disease vector control programs among eleven countries and subnational states in South Asia and the Middle East. In each country or state, independent teams conducted vector control needs assessment with engagement of stakeholders. Important weaknesses were described for malaria, dengue and leishmaniases regarding vector surveillance, insecticide susceptibility testing, monitoring and evaluation of operations, entomological capacity and laboratory infrastructure. In addition, community mobilization and intersectoral collaboration showed important gaps. Countries and states expressed concern about insecticide resistance that could reduce the continued effectiveness of interventions,

**Data Availability Statement:** The data contained in the reports on which this work was based cannot be shared because they are property of third parties and the informed consent did not include the permission. The reports can be requested from the third-party sources listed in the S2 Appendix by researchers who meet the criteria for access to confidential data.

**Funding:** The work was supported partly by the Bill & Melinda Gates Foundation, Seattle, WA, USA under the Innovation-2-Impact project (grant number OPP1133139) awarded to the World Health Organization (RV and RSY), and partly by the WHO Global Malaria Programme (internal funds). The former funder had no role in design, data collection and analysis, decision to publish, or preparation of the manuscript.

**Competing interests:** The authors have declared that no competing interests exist.

which demands improved monitoring. Moreover, attainment of disease elimination necessitates enhanced vector surveillance. Vector control needs assessment provided a useful planning tool for systematic strengthening of vector control systems. A limitation in conducting the vector control needs assessment was that it is time- and resource-intensive. To increase the feasibility and utility of national assessments, an abridged version of the guidance should focus on operationally relevant topics of the assessment. Similar reviews are needed in other regions with different contextual conditions.

## Author summary

Vector control can play a major role in the control and elimination of vector-borne diseases, such as malaria, dengue, leishmaniases and other vector-borne diseases. However, to reach its potential, national vector control systems should be adequately supported by vector surveillance, community participation and intersectoral collaboration. As a step towards strengthening vector control, it has been advocated that countries conduct a vector-control needs assessment. The authors reviewed the needs or gaps as perceived by stakeholders of disease vector control programs in eleven countries and subnational states in South Asia and the Middle East. Programs for control and elimination of malaria, dengue and leishmaniases had major shortcomings in vector surveillance and entomological capacity. This was a concern because vectors develop insecticide resistance which, if unchecked, could reduce the effectiveness of interventions. Also, attaining elimination of disease demands enhanced vector surveillance support. There were major gaps in community mobilization and intersectoral collaboration. The findings imply that vector control systems should adapt to the changing disease situation and adopt a cross-disease mandate. The authors propose improvements to the methods of needs assessment.

## Introduction

The global burden caused by vector-borne diseases such as malaria, dengue, lymphatic filariasis and leishmaniases is unacceptably high [1], and people in many parts of the World are at risk of two or more vector-borne diseases [2]. Despite the progress made in some regions with the control and elimination of malaria, visceral leishmaniasis and lymphatic filariasis [3–5], arboviral diseases such as dengue, chikungunya and Zika virus have recently expanded [6–8]. The transmission of vector-borne disease pathogens can be suppressed or interrupted by vector control interventions, and vector control has had a demonstrated impact on several vector-borne diseases [9–11]. Vector control tools that have caused dramatic reductions in malaria, and in some cases also in leishmaniases, are insecticide-treated nets (ITNs) and indoor residual spraying (IRS) [12–14]. Another valuable vector control tool is larval source management, which includes source reduction and larviciding, and which can be effective against malaria and dengue in specific settings [15,16].

Contemporary challenges in disease vector control include inadequate political and financial commitment; societal and environmental changes, including unplanned urbanization; emerging diseases and invasive vectors; and the development of insecticide resistance. To reach its full potential, vector control should be supported by a vector surveillance system with adequate entomological expertise, with sufficient participation of communities and collaboration between programs and sectors [10,17]. The Global Vector Control Response 2017–2030, which

is based on the concepts of integrated vector management, calls on countries and development partners to strengthen vector control as a fundamental approach to preventing disease and responding to outbreaks [18,19]. The first activity among those prioritized for achieving the targets of the Global Vector Control Response is to have a vector control needs assessment conducted and resource mobilization plans developed by countries. Hence, the vector control needs assessment is seen as a starting point for strengthening national vector control systems.

Needs assessment is a form of strategic planning that is occupied with finding the gaps, called needs, between what currently is and what should ideally be, and that prioritizes those needs to guide decisions about what to do next [20]. Needs assessment thus involves establishing the current situation, the desired situation, and the discrepancy between them. The process of vector control needs assessment was first developed in 2003 at regional level as a tool for understanding current strengths and weaknesses of vector control systems, and thus for programs and governments to take steps to enact improvements. In 2017, WHO published a guidance document for a national vector control needs assessment [21], which was aligned with the framework of the Global Vector Control Response. Vector control needs assessment has been implemented in several countries across regions [22]. However, there has not been a review of the outputs of the vector control needs assessments. Our objective was to review the perceived weaknesses in vector control systems that countries are confronted with, and whether the vector control needs assessment is effective in the identification and prioritization of those weaknesses.

## Methods

### Selection of countries and states

The WHO South-East Asia and Eastern Mediterranean Regions were selected for this review as regions in which vector control programs have been implemented against vector-borne diseases and as the only regions for which assessment reports were available. Only reports which were based on the 2017 WHO guidance document for a national vector control needs assessment [21] were eligible for selection. Thirteen eligible reports were available; eleven of these were from the subregions of South Asia and the Middle East, and consequently, these subregions were set as the geographic scope of our review; for this reason, the available reports from Morocco and Sudan were excluded. The eleven eligible reports were from Bangladesh, Assam (India), Gujarat (India), Jharkhand (India), Tamil Nadu (India), Islamic Republic of Iran, Iraq, Maldives, Nepal, Sri Lanka, and Yemen. The assessments were conducted at national level except for India, where assessments were conducted at subnational (i.e., federated state) level, because of this country's large size and population. The four Indian states representing different eco-epidemiological settings and health system capacities of the country had previously been selected by the WHO Regional Office in New Delhi for conducting the needs assessments: (i) Assam, north-eastern India, forest-hill ecosystem with tea gardens and rice agriculture, and with deficient health system capacity; (ii) Gujarat, western India, semi-arid, urban and irrigated plains ecologies, and with a well-developed health system capacity; (iii) Jharkhand, eastern India, forest-tribal dominated hilly ecology, poor socio-economic conditions, with a partly-deficient health system capacity; and (iv) Tamil Nadu, southern India, irrigated rice agriculture and urban settings, with a more advanced health system capacity.

### Procedure

The WHO Regional Offices in New Delhi and Cairo requested WHO Country Offices in the selected countries to engage national consultants in each country/state, with one or two national consultants per country/state. The national consultants were selected based on their vector control expertise and their familiarity with the national or subnational systems

of vector control. The assessments were conducted in 2018–2021, depending on the country/state (Table 1). The tool used by the national consultants for their assessment was the WHO guidance document for a national vector control needs assessment, particularly the questionnaire in its annex [21]. The national consultants engaged with a panel of stakeholders in each country/state through interviews and consultation meetings (S1 Appendix) and conducted a desk review of relevant reports and publications. The results of this process were documented by the national consultants in a report from each country/state. Hence, eleven reports were produced by the selected countries/states, which were submitted via WHO country offices to the authors (S2 Appendix). As a follow-up to the reports, we solicited responses to our specific queries to verify or supplement the information provided in the reports. We also consulted available national strategic plans for malaria, visceral leishmaniasis, dengue and lymphatic filariasis from Bangladesh, India, Sri Lanka, and Nepal.

### Analytical framework

We developed an analytical framework, based on available logic models [23], for evaluating the process and results of the vector control needs assessment. The framework consisted of a

**Table 1. Details per country/state regarding diseases for which a vector control component existed.**

| Country/state | Population (mln)[a] | Disease | Disease form | Disease deaths[b] | Disease cases[b] | 5-yr change in disease cases (%)[b] | Intervention type[c] | Assessment year[d] |
|---|---|---|---|---|---|---|---|---|
| Bangladesh | 166 | Malaria | | 9 | 17225 | -70 | 2,3,4 | 2020 |
| | | Dengue | | 179 | 101354 | >+1000 | 1,2,5 | 2020 |
| | | Leishmaniasis | Visceral | 5 | 216 | -80 | 4 | 2020 |
| India/Assam | 31 | Malaria | | n/a | 1459 | -87 | 3,4 | 2020 |
| | | Dengue | | 0 | 196 | +131 | 1,2,5 | 2020 |
| India/Gujarat | 60 | Malaria | | 1 | 13883 | -66 | 2,3,4,5 | 2020 |
| | | Dengue | | 17 | 18455 | +230 | 1,2,5 | 2020 |
| India/ Jharkhand | 33 | Malaria | | 2 | 37063 | -64 | 3,4 | 2020 |
| | | Dengue | | 1 | 463 | >+1000 | 1,2 | 2020 |
| | | Leishmaniasis | Visceral | 0 | 541 | -57 | 4 | 2020 |
| India/Tamil Nadu | 72 | Malaria | | n/a | 2088 | -63 | 2,3,4 | 2020 |
| | | Dengue | | 5 | 8527 | +237 | 1,2,5 | 2020 |
| Iran | 87 | Malaria | | 0 | 0 | -∞ | 2,3,4,5 | 2018 |
| | | Leishmaniasis | Cutaneous / visceral | n/a | 8161 / 72 | -49 / +177 | 6 | 2018 |
| Iraq | 44 | Leishmaniasis | Cutaneous / visceral | n/a | 7056 / 170 | +162 / -53 | 3,4,6 | 2018 |
| Maldives | 1 | Dengue | | 2 | 5022 | n/a | 1,2,5 | 2020 |
| Nepal | 29 | Malaria | | 0 | 464 | -91 | 3,4 | 2021 |
| | | Dengue | | 6 | 14662 | n/a | 1 | 2021 |
| | | Leishmaniasis | Visceral | n/a | 185 | -41 | 4 | 2021 |
| Sri Lanka | 22 | Malaria | | 0 | 0 | 0 | 2,3,4 | 2020 |
| | | Dengue | | n/a | 105049 | +121 | 1,2,4,5 | 2020 |
| Yemen | 32 | Malaria | | 2108 | 831533 | +42 | 3,4 | 2018 |

[a] Total population; data sources: Federated states of India [26]; other countries [27].

[b] Annual number of disease deaths and disease cases in 2019, and the change in disease cases over the period 2014–2019 (see Methods). N/a, not available.

[c] Vector control interventions: 1, Source reduction or environmental management; 2, larviciding; 3, insecticide-treated nets; 4, indoor residual spraying; 5, space spraying (fogging); 6, rodent control.

[d] Year of completion of the vector control needs assessment.

'results chain' and a comparison between the current and desired situation at each stage of the results chain (Fig 1). The results chain shows the flow from inputs and activities (the 'means') of a program to outputs, outcomes, and impact (collectively the 'results'). The topics pertaining to each step in the results chain were aligned with the framework of the Global Vector Control Response [18]. Programs require input in terms of capacity building, research, human resources, financial resources, and infrastructure. These inputs are used for four types of activities of (i) scaling up and integration of interventions; (ii) surveillance and monitoring & evaluation (M&E); (iii) community mobilization; and (iv) intra/intersectoral collaboration. The inputs and activities are influenced by the enabling factors, most directly by the availability of national strategic plans, guidelines, and organizational structures. The activities give rise to outcomes, such as reduced vector populations or reduced pathogen transmission rates. Finally, the outcomes contribute to impact in terms of a reduction in disease incidence.

Needs assessment starts from what we want to achieve in terms of impact. The needs are the gaps in the performance or conditions of a program, which is, the gaps between the current situation and the desired situation [24]. When a gap is identified between the current impact and the desired impact, this could expose possible gaps lower down the results chain, in terms of outcomes, activities, inputs and enabling factors of a program [23]. For example, the gap between the current disease situation and the desired disease situation, as defined in national strategic plans, could be caused by ineffective vector control, poor coverage with interventions, or poor compliance by communities, which may be attributable to inadequate inputs or lack of a strategic plan.

## Data processing and analysis

For each country/state, the diseases with a programmatic vector control component were identified. Diseases for which a program or a vector control component was lacking in a country/state were omitted from the analysis for that country/state. For each disease, we took the annual number of deaths and indigenous cases in 2019. We also took the five-year change (%) in disease cases since 2014, as an indication of programmatic impact on disease in recent years at country or state level; for Jharkhand, dengue case data were available for the period 2013–2018. The data on deaths and cases were obtained from the reports and follow-up consultation. Gaps in data on deaths and cases were supplemented, where needed, from other sources,

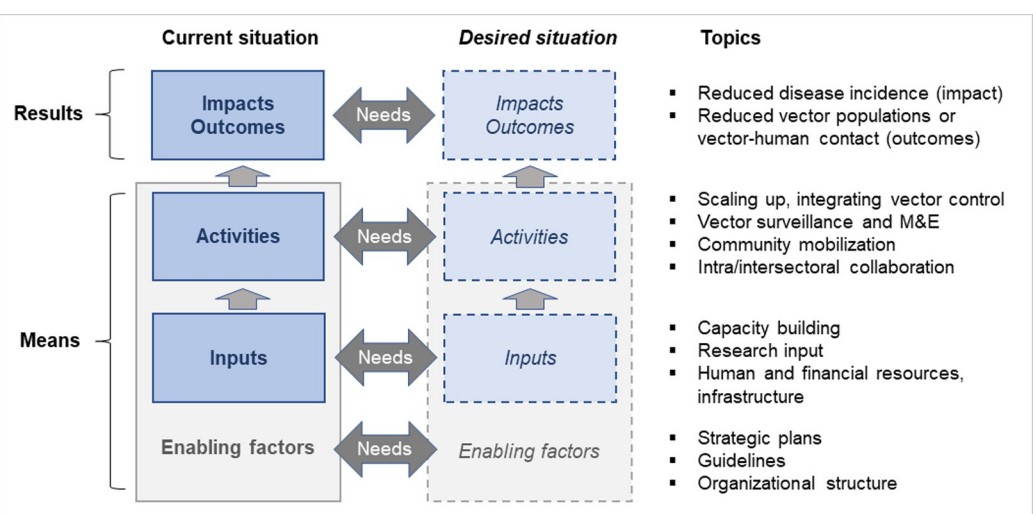

**Fig 1. Analytical framework for vector control needs assessment, based on available logic models, and aligned to the Global Vector Control Response 2017–2030.**

specifically malaria case data from Iran, Nepal and Yemen [3], and leishmaniases case data from Iran, Iraq and Nepal [25]. In post-elimination settings, we refer to indigenous cases only.

Categories and key items of a vector control system were construed according to the analytical framework (see descriptions in the S1 Table) and were listed by country/state and disease in a matrix format. The perceptions about the situation and needs of the vector control systems by the national consultants and stakeholders were extracted and interpreted from each report by one of us (HvdB). Based on the narratives in the reports, we appraised and categorized the perceived needs regarding each key item as (a) 'adequate' (i.e., no immediate need for improvement), (b) 'partly adequate' (i.e., basic item in place but need to be upgraded or updated), (c) 'inadequate or absent' (i.e., clear need for improvement), or no information provided which we processed as missing data. These categorized results were used to populate the matrix to provide an overview of the perceived needs per disease per country/state. We then reviewed the results by comparing the key items across countries/states, and between diseases, and by examining the full set of key items per country/state.

A pooled analysis of the perceptions across countries/states was conducted per category and per disease. The purpose of this analysis was to study the average and variability of results in the set of (sub)national assessments, while noting that substantial differences in size, population and vector control conditions exist between countries/states. Numeric values of +1, 0, and -1 were assigned to the categorization as 'adequate', 'partly adequate' and 'inadequate or absent', respectively. For each disease (malaria, dengue, and leishmaniases), average numeric values of key items per category were synthesized, first, per country/state and then, across countries/states, with standard error given as an estimate of variability between countries/states. The numerical data used in Fig 2 are included in the S1 Data.

## Results

We present the results from the eleven countries/states as follows. We first discuss the state of the art of vector control for each disease per country/state. We then present the results of the (sub)national assessments on perceptions regarding malaria, dengue and leishmaniases in

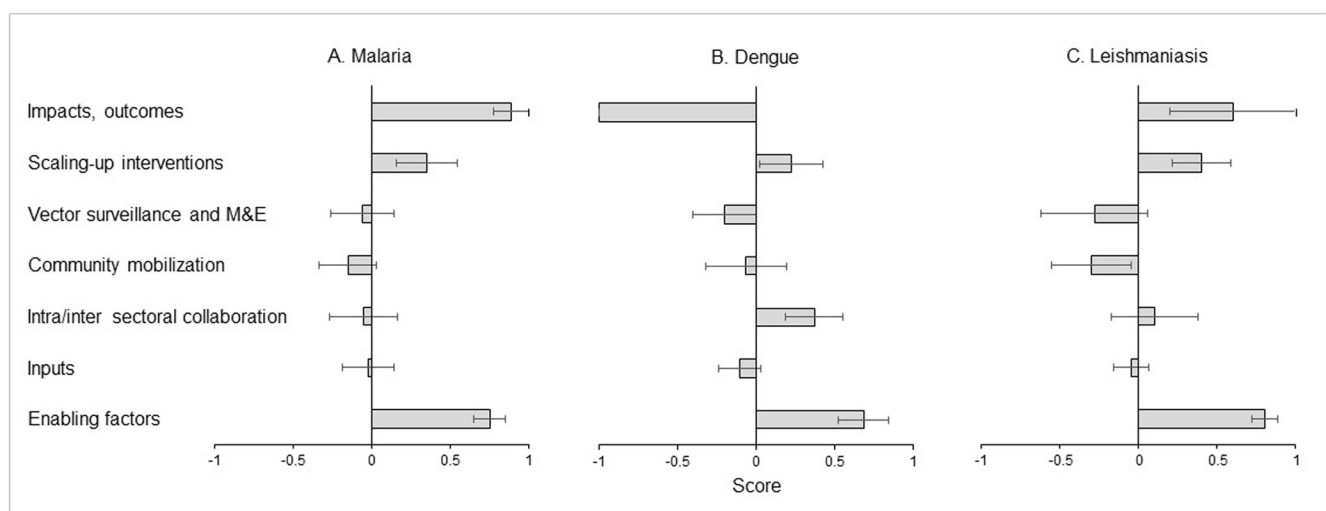

**Fig 2. Synthesis of the perceptions of vector control systems for malaria, dengue and leishmaniases (A-C). Presented is the average score with standard error per category across countries/states (n = 9, 8 and 5 countries/states for malaria, dengue and leishmaniases, respectively), as explained in the Methods. A score of 1 indicates 'adequate'; -1 indicates inadequate.**

separate sections on (a) impact and outcomes, (b) activities, (c) inputs, and (d) enabling factors. Finally, we provide a synthesis of results across countries/states.

## Vector control state of the art

Among the eleven selected countries/states, a programmatic vector control component was existent in nine countries/states for malaria; in eight for dengue; and in five for cutaneous and visceral leishmaniases (Table 1). The number of country/states with an existent vector control component for one, two, and three diseases was three, five, and three country/states, respectively. The main vector control intervention types used were ITNs, IRS and larviciding against malaria; source reduction or environmental management, larviciding and space spraying against dengue; and IRS, ITNs and rodent control against leishmaniases (Table 1).

Data on the use of vector control insecticide products were mostly absent from the assessment reports but a recent WHO study found that vector control insecticide use in the Asia-Pacific Region consisted mostly of pyrethroids and organochlorines for the control of malaria and leishmaniasis, and pyrethroids and organophosphates for the control of dengue [28]. Several countries/states expressed concern about the development of insecticide resistance in disease vectors of malaria, dengue and leishmaniases, but data on insecticide susceptibility testing were not systematically reported.

No separate vector control component existed for chikungunya while several reports indicated that vector control for chikungunya was integrated with that for dengue; these viruses are transmitted by the same aedine vector species. Control or elimination of Japanese encephalitis and lymphatic filariasis relied on targeted vaccination, and mass drug administration with case management, respectively, but did not generally include a programmatic vector control component. Exceptions were occasional space spraying operations in villages with positive case reporting of Japanese encephalitis in Tamil Nadu; and larviciding, sanitation and space spraying in urban areas for control of nuisance mosquitoes which include lymphatic filariasis vectors in three countries/states. In Iran, tick control measures were carried out in response to sporadic human cases of Crimean-Congo haemorrhagic fever; however, these measures were implemented by the agriculture ministry, not the health ministry. In Yemen, malaria, dengue, schistosomiasis, leishmaniasis, and onchocerciasis were all endemic, but details about a programmatic vector control component were only reported for malaria, except that it was mentioned that fogging and chemical larviciding have been used in response to dengue outbreaks. The reports from Nepal and Tamil Nadu called for future attention to mite- and tick-borne diseases, for control of scrub typhus (Nepal; Tamil Nadu) and Kyasanur forest disease (Tamil Nadu). Hence, in some countries/states, the absence of vector control for certain diseases could signify the need for establishing a vector control component.

## (Sub)national assessments

**Impact and outcomes.**   Starting from the impact level, it is important to know whether the perceptions of impact and outcomes of the operations have been adequate or not, when compared with targets in the available strategic plans. However, none of the countries or states compared the disease situation and trends with the milestones and targets outlined in previous disease-specific strategic plans. Instead, we extracted the perceptions of programmatic impact on disease from the narratives in the reports. The impact on malaria was perceived as 'adequate' in eight countries/states (Table 2), as supported by the 63–91% reductions in malaria cases recorded over the period 2014–2019 (Table 1). An exception was Yemen, where an increase in malaria cases in recent years was associated with the severe implications of the civil conflict on the functions of the health sector. Iran achieved zero indigenous malaria cases since 2018. Sri

**Table 2. Overview of perceptions about the system for malaria vector control.[a]**

| Key item[b] | Bangladesh | India/ Assam | India/ Gujarat | India/ Jharkhand | India/ Tamil Nadu | Iran | Nepal | Sri Lanka | Yemen |
|---|---|---|---|---|---|---|---|---|---|
| **IMPACT & OUTCOMES:** | | | | | | | | | |
| Disease impact | green | green | green | green | green | green | green | green | (blank) |
| Vector outcomes | (blank) | (blank) | (blank) | (blank) | (blank) | (blank) | (blank) | (blank) | (blank) |
| **ACTIVITIES:** | | | | | | | | | |
| *a. Scaling-up, integrating vector control:* | | | | | | | | | |
| Coverage | green | orange | green | green | green | green | orange | green | orange |
| Operational quality | (blank) | (blank) | (blank) | (blank) | (blank) | green | (blank) | (blank) | (blank) |
| Pesticide management | red | green | green | green | green | orange | orange | green | red |
| *b. Vector surveillance and M&E:* | | | | | | | | | |
| Routine entomological surveys | red | orange | orange | red | orange | green | red | green | orange |
| Routine insecticide susceptibility tests | red | orange | orange | orange | green | green | red | green | green |
| Sentinel sites | orange | green | green | orange | green | green | red | green | orange |
| M&E of operations | (blank) | red | red | green | orange | green | red | green | (blank) |
| Data management and sharing | (blank) | green | green | green | orange | green | red | green | orange |
| *c. Community mobilization:* | | | | | | | | | |
| Systematic implementation | orange | orange | green | orange | red | green | orange | orange | green |
| Monitoring | red | red | (blank) | (blank) | red | orange | red | orange | red |
| *d. Intra/intersectoral collaboration:* | | | | | | | | | |
| Cross-disease collaboration | red | green | green | green | green | orange | green | orange | red |
| Intersectoral committee | red | green | green | red | red | green | green | red | red |
| Intersectoral action | red | green | (blank) | red | green | green | red | (blank) | red |
| **INPUTS:** | | | | | | | | | |
| Entomologists | orange | red | red | red | orange | green | orange | green | green |
| Vector control operators | green | green | green | red | green | green | green | green | green |
| Training, entomology | orange | red | orange | (blank) | red | green | red | green | green |
| Training, operations | orange | (blank) | green | green | red | green | (blank) | green | green |
| Entomology laboratories | orange | red | green | red | orange | green | orange | orange | green |
| Research input | orange | red | green | green | (blank) | orange | orange | orange | green |
| **ENABLING FACTORS:** | | | | | | | | | |
| Strategic plans | green | green | green | green | green | green | green | green | green |
| Guidelines | orange | green | green | green | green | green | green | green | (blank) |
| Organizational structure | green | (blank) | green | (blank) | green | green | orange | green | orange |

[a] Perceptions, as described in the text, are categorized as (a) adequate, shaded green; (b) partly adequate, shaded orange (e.g., need to be upgraded or updated); (c) inadequate or absent, shaded red; or (d) no information provided, no shading.

[b] Key items are described in the S1 Table.

Lanka was certified malaria-free in 2016 and maintained zero indigenous malaria cases in recent years, indicating that the prevention of re-establishment had been effective.

The impact of vector control interventions on dengue was perceived as 'inadequate' in all eight countries/states with a dengue vector control component (Table 3), as demonstrated by the sharp increases in dengue cases over the previous five years (Table 1). The impact on leishmaniases was perceived as 'adequate' in four countries/states and as 'inadequate' in one country with a leishmaniases vector control component (Table 4); the leishmaniases cases over the period 2014–2019 showed marked reductions in most instances, but Iraq and Iran, where both

**Table 3. Overview of perceptions about the system for dengue vector control.[a]**

| CATEGORY / Key item[b] | Bangladesh | India/Assam | India/Gujarat | India/Jharkhand | India/Tamil Nadu | Maldives | Nepal | Sri Lanka |
|---|---|---|---|---|---|---|---|---|
| **IMPACT & OUTCOMES:** | | | | | | | | |
| Impact on disease | red | red | red | red | red | red | red | red |
| Vector outcomes | | | | | | | | |
| **ACTIVITIES:** | | | | | | | | |
| *a. Scaling-up, integrating vector control:* | | | | | | | | |
| Coverage | orange | orange | orange | | orange | orange | | green |
| Operational quality | | | | | | red | | |
| Pesticide management | red | green | green | green | green | red | orange | orange |
| *b. Vector surveillance and M&E:* | | | | | | | | |
| Routine entomological surveys | red | green | orange | red | green | red | red | orange |
| Routine insecticide susceptibility tests | red | orange | orange | orange | | red | red | red |
| Sentinel sites | orange | green | green | red | green | orange | orange | orange |
| M&E of operations | | red | red | green | orange | orange | red | orange |
| Data management and sharing | | green | green | green | orange | orange | red | green |
| *c. Community mobilization:* | | | | | | | | |
| Systematic implementation | orange | green | green | orange | red | orange | orange | green |
| Monitoring | red | red | | | red | orange | orange | |
| *d. Intra/intersectoral collaboration:* | | | | | | | | |
| Cross-disease collaboration | red | green | green | green | green | green | green | orange |
| Intersectoral committee | green | | green | | red | orange | green | green |
| Intersectoral action | red | | | red | red | orange | red | green |
| **INPUTS:** | | | | | | | | |
| Entomologists | orange | | red | | orange | orange | red | orange |
| Vector control operators | green | green | green | green | orange | red | green | green |
| Training, entomology | red | | orange | | orange | red | | orange |
| Training, operations | orange | | green | orange | orange | orange | | green |
| Entomology laboratories | red | red | red | red | red | red | orange | green |
| Research input | orange | | | orange | | red | | orange |
| **ENABLING FACTORS:** | | | | | | | | |
| Strategic plans | red | green | green | green | green | red | | green |
| Guidelines | orange | green | green | green | green | green | green | green |
| Organizational structure | green | | green | | green | orange | orange | green |

[a] Perceptions, as described in the text, are categorized as (a) adequate, shaded green; (b) partly adequate, shaded orange; (c) inadequate or absent, shaded red; or (d) no information provided, no shading.

[b] Key items are described in the S1 Table.

forms of leishmaniases were present, showed recent increases in cutaneous and visceral leishmaniasis, respectively (Table 1).

Outcomes of vector control operations typically include changes in vector densities, biting rates, or transmission rates that are not due to seasonality. Outcome assessment can enable the evaluation of whether vector control has or has not been effective. None of the countries or states reported that such entomological outcomes were available or that they were used for evaluation of vector control for malaria, dengue or leishmaniases, suggesting this issue requires further attention.

**Table 4. Overview of perceptions about the system for leishmaniases vector control.[a]**

| CATEGORY<br>Key item[b] | Bangladesh | India/ Jharkhand | Iran | Iraq | Nepal |
|---|---|---|---|---|---|
| **IMPACT & OUTCOMES:** | | | | | |
| Impact on disease | green | green | green | red | green |
| Vector outcomes | | | | | |
| **ACTIVITIES:** | | | | | |
| *a. Scaling-up, integrating vector control:* | | | | | |
| Coverage | green | green | | green | orange |
| Operational quality | | | green | | |
| Pesticide management | red | green | orange | orange | orange |
| *b. Vector surveillance and M&E:* | | | | | |
| Routine entomological surveys | red | red | green | | red |
| Routine insecticide susceptibility tests | red | orange | green | orange | red |
| Sentinel sites | red | orange | red | green | red |
| M&E of operations | | green | | green | red |
| Data management and sharing | | orange | orange | green | red |
| *c. Community mobilization:* | | | | | |
| Systematic implementation | orange | orange | green | red | orange |
| Monitoring | red | | orange | | red |
| *d. Intra/intersectoral collaboration:* | | | | | |
| Cross-disease collaboration | red | green | orange | | green |
| Intersectoral committee | red | green | green | green | green |
| Intersectoral action | red | red | | red | red |
| **INPUTS:** | | | | | |
| Entomologists | orange | orange | orange | orange | red |
| Vector control operators | green | red | orange | orange | green |
| Training, entomology | red | | orange | orange | red |
| Training, operations | orange | green | orange | green | |
| Entomology laboratories | orange | red | green | green | orange |
| Research input | orange | orange | orange | red | orange |
| **ENABLING FACTORS:** | | | | | |
| Strategic plans | green | green | green | | green |
| Guidelines | orange | green | green | | green |
| Organizational structure | green | | orange | green | orange |

[a] Perceptions, as described in the text, are categorized as (a) adequate, shaded green; (b) partly adequate, shaded orange; (c) inadequate or absent, shaded red; or (d) no information provided, no shading.

[b] Key items are described in the S1 Table.

**Activities.** a. Scaling-up, integrating vector control: The reports provided few details regarding the implementation of vector control, the quality of interventions, timing, stratification and targeting of interventions, and whether implementation was in accordance with national strategic and operational plans. The coverage of populations with vector control interventions was perceived as 'adequate' or 'partly adequate' for malaria and leishmaniases but was considered only 'partly adequate' for dengue. Operational quality was not reported on, except by Iran and the Maldives. With regard to pesticide management, the four Indian states reported that national guidelines on pesticide management were followed by the programs;

the seven other countries described shortcomings in pesticide management regarding training on safe handling, personal protection, safe storage, or regulation.

b. Vector surveillance and M&E: Critical shortcomings were reported regarding routine entomological surveys for malaria, with major differences reported between countries/states (Table 2). In most cases, surveillance of malaria vectors was patchy or irregular due to shortcomings in entomological capacity, including entomologists, technicians, and equipment; or surveillance covered only few entomological parameters. Exceptions were Sri Lanka and Iran where malaria vector surveillance was appraised as 'adequate'. Routine insecticide susceptibility testing of malaria vectors was perceived as 'inadequate' or 'partly adequate' in seven out of nine malaria programs (Table 2). Similarly, inadequacies were reported for the presence of sentinel sites, which are needed for monitoring temporal changes in entomological parameters. The monitoring and evaluation (M&E) of malaria vector control operations was perceived as 'inadequate' in three countries/states. Five out of eight countries/states reported adequate data management and data sharing on malaria vector surveillance and control. For dengue and leishmaniases, the perceived situation on vector surveillance and M&E was similar to malaria, except in Sri Lanka and Iran, where it was perceived as inferior to that for malaria (Tables 3 and 4). In none of the countries/states, routine insecticide susceptibility testing of dengue or leishmaniases vectors was considered as 'adequate'. It was noted from Tamil Nadu that vector surveillance was particularly weak in urban centers, where most malaria and dengue cases occurred.

c. Community mobilization: Systematic implementation of community mobilization was considered to be 'adequate' in only a minority of the programs on malaria, dengue and leishmaniases (Tables 2–4). Reported community mobilization activities included campaigns, school rallies, house-to-house visits, and use of media, intended to promote disease prevention, health-seeking behavior, and compliance with interventions. In most programs on malaria, dengue and leishmaniases, the monitoring of community mobilization was reported to be 'absent' or 'partly adequate', or no information was provided (Tables 2–4).

d. Intra- and intersectoral collaboration: The collaboration on vector control between disease control programs was perceived as 'adequate' in the four Indian states because all vector-borne diseases were operated under one national umbrella program with a cross-disease mandate (Tables 2–4). In Nepal and Iraq, a single vector control unit was present at national level to oversee all vector-borne diseases. In Sri Lanka, the malaria program and the dengue program collaborated on vector surveillance and vector control at the district level. Lack of coordination between vector control programs was reported from Bangladesh and Yemen.

Establishment of an intersectoral committee or task force can facilitate multisectoral engagement in vector control across diseases [18]. Functional committees were 'adequate' in less than half of the malaria programs but were more common in dengue programs because the committees in Bangladesh and Sri Lanka focused on dengue only (Tables 2–4). Intersectoral action on vector control was perceived as 'inadequate' or 'absent' in most programs (Tables 2–4). Hence, the presence of a committee was no assurance for intersectoral action, as was apparent in Bangladesh, Jharkhand, Iraq, and Nepal. In Iran, the malaria program collaborated with a provincial water and wastewater company to repair broken water pipes to contain breeding sites of malaria vectors. In Sri Lanka, cleaning campaigns for dengue control involved collaboration of the dengue program at decentralized level with the police, armed forces, local government, and community leaders.

**Inputs.**   As inputs, we considered human resources, in terms of entomologists, and vector control operators (spray workers; supervisors); capacity building, in terms of training on entomology and operations; infrastructure, in terms of entomology laboratories; and research input. Financial inputs were excluded from our evaluation. Entomological capacity and expertise were perceived as 'inadequate' or 'partly adequate' in nearly all malaria programs, and in

all dengue and leishmaniases programs (Tables 2–4). Exceptions were Sri Lanka and Yemen, where the perception was that adequate entomologists were available for prevention and control of malaria. The shortage of entomologists in the four Indian states and in Bangladesh was primarily caused by high levels of vacancies. The reports of Bangladesh and Gujarat specified the lack of promotional avenues for entomologists as a possible cause of vacancies. Vector control operators were perceived as 'adequate' or 'partly adequate' in most programs.

Training in entomology, including in vector surveillance and decision making on vector control, was perceived as 'inadequate' or 'partly adequate' in all but one malaria program and in all dengue and leishmaniases programs. An exception was Sri Lanka, where training requirements for malaria entomology were adequately adjusted to the prevention of re-establishment phase. Training on operations also showed shortcomings in several programs (Tables 2–4). Entomology laboratories, which are needed for processing of field samples, insecticide susceptibility testing, and operational studies, were perceived as 'adequate' in three out of nine malaria programs, one out of eight dengue programs, and two out of five leishmaniases programs (Tables 2–4).

Research input for vector control was perceived as 'inadequate' or 'partly adequate' in all but one program (Tables 2–4). Weaknesses were reported in the coordination or prioritization of research for operational relevance, and the lack of data sharing between research and operations. The report from Tamil Nadu highlighted the specific need for research on integrated control of *Anopheles stephensi* and *Aedes aegypti* in urban environments, and research on control options for neglected but emerging mite- and tick-borne diseases, such as scrub typhus and Kyasanur forest disease.

**Enabling factors.** We reviewed the availability of national strategic plans, national guidelines, and organizational structures as enabling factors for vector surveillance and vector control. National strategic plans were 'adequate' for all malaria programs, five out of eight dengue programs and four out of five leishmaniases programs. National guidelines on vector control were 'adequate' in most programs on malaria, dengue and leishmaniases.

The reports from most countries/states noted that an 'adequate' or 'partly adequate' organizational structure was in place defining roles, responsibilities, and lines of reporting on vector control. Most dengue programs were decentralized to the municipalities and districts, but the Maldives noted the roles and responsibilities regarding vector control had not been defined at decentralized level, whilst in Nepal, recent federalization of the government system negatively affected the vector surveillance and control activities.

The reports elaborated on policies that might support or conflict with disease vector control. However, this rich and complex information was not amenable to be categorized as 'adequate', 'partly adequate' or 'inadequate'. All countries/states listed policies available in other sectors with relevance for vector control. Examples are a policy on solid waste management, policy on pollution abatement in rivers and lakes, policy on environmental impact assessment of new projects, policy on integrated pest management in agriculture, regulation on prevention of mosquito breeding, and policy on international health regulations at points of entry.

## Synthesis

The analysis of the perceived situation captured the results per category across countries/states (Fig 2). The impact on disease was perceived as 'adequate' for malaria and leishmaniases, indicated by a score approaching the value of 1, but 'inadequate' for dengue, indicated by a score of -1. Scaling-up of interventions was perceived as moderately adequate (0.23–0.46) for the three diseases, but with substantial variation between countries/states. The score for vector surveillance and M&E was negative for all diseases, especially for dengue and leishmaniases, indicating it was

'inadequate' or 'partly adequate'. The scores for community mobilization were also marginally negative for all diseases, whilst the score for intra/intersectoral collaboration was marginally negative for malaria, neutral for leishmaniases, and positive for dengue. The score for inputs, in terms of human resources, capacity building, infrastructure and research, was neutral or marginally negative for the three diseases. The enabling factors, in terms of suitable strategies, guidelines and organizational structure, scored distinctly positive for the three diseases.

Most of the reports contained a concluding section in which what were conceived as priority needs were listed, the most frequent of which referred to the improvement of vector surveillance and insecticide resistance monitoring; the strengthening of entomological expertise, equipment and laboratory infrastructure; the establishment of an intersectoral committee on integrated vector management; and enhancement of research inputs.

The reports also discussed contemporary or future challenges to vector control. First, concern about the development of insecticide resistance in disease vectors of malaria, dengue and leishmaniases was expressed in the reports from Gujarat, Jharkhand, Bangladesh and Sri Lanka. The second challenge was to attain and maintain disease elimination. For example, the report from Tamil Nadu highlighted the need for enhanced vector surveillance owing to the role of the partly outdoor biting behavior of *An. culicifacies* and *An. stephensi* in maintaining residual malaria transmission, including in urban centers. In the four Indian states, filling the vacant entomology posts was deemed necessary for eliminating malaria. It was reported from Sri Lanka that a financial transition has been adopted to ensure that the malaria-free status is sustained. A third challenge was reported regarding the emergence of vector-borne diseases and invasive vector species, including the geographic expansion, and increasing incidence of arboviral diseases in the subregions.

## Discussion

National or subnational systems of disease vector control comprise technical, logistical, and institutional elements for the control, elimination or prevention of multiple vector-borne diseases. Understanding the functioning of these systems within dynamic epidemiological, socioeconomic, and environmental settings is vital to ensure cost-effective, efficient, and adaptive implementation of vector control. The WHO guidance document for a national vector control needs assessment was intended as a tool to help countries analyze their vector control systems and propose structural improvements [21]. Evidently, use of the tool through multistakeholder consultations enabled the selected countries/states to identify critical shortcomings and needs. The results disclosed a mixed picture of the situation of vector control systems, with large differences between countries/states. Important weaknesses were described for malaria, dengue and leishmaniases regarding vector surveillance, insecticide susceptibility testing, M&E of operations, entomological expertise and laboratory infrastructure. In addition, community mobilization and intersectoral collaboration, which have been advocated as pillars of action in the Global Vector Control Response [18], exposed important gaps. There has been a shortage of international resources to support national vector control systems [29]. Substantial investment will be needed in most of the countries/states to strengthen their capacity for vector surveillance and insecticide resistance monitoring using domestic and international funding sources.

Entomological capacity was identified as a scarce asset, and in some countries, the available capacity was disease-specific and not shared for the control of other vector-borne diseases. Recruitment of entomologists, training and strengthening of laboratory infrastructure are urgently needed as inputs for strengthening the vector control systems. Moreover, the collaboration between vector control programs through establishment of a vector control unit with cross-disease mandate has potential to improve efficiency through the sharing of entomological

expertise, capacity and vector control resources [30], considering that some diseases may be eliminated while others are emerging.

Despite the positive programmatic impact on malaria and visceral leishmaniasis with the use of ITN and IRS interventions, countries/states expressed concern about contemporary and future challenges that could reduce the continued effectiveness of these interventions and, thus, undermine efforts to attain national targets for disease control and elimination. Vectors can adapt to the use of ITNs and IRS by developing insecticide resistance, behavioral resistance, or by changing their species composition [31–33]; hence, strong vector surveillance is essential for timely detection of such changes to trigger an appropriate response regarding the selection and use of vector control tools and products.

In dengue control, an additional challenge has been the lack of effective control tools, especially in the absence of medications or vaccines that tackle the reservoir of dengue virus. Most of the available dengue vector control tools have been developed in an era when epidemiological outcomes were not emphasized in the assessment of efficacy [34]. Moreover, it has proven difficult in urbanized areas to achieve and sustain high coverage of breeding sites with source reduction and larviciding over the length of the transmission season [35]; this was reflected in our review by shortcomings in community mobilization and operational coverage of dengue vector control.

Another challenge faced by the countries was to attain or maintain disease elimination. When transmission nears zero, intensified epidemiological and entomological surveillance is needed for fine-grained stratification of at-risk areas, with identification of hotspots, to improve the targeting of operations. Moreover, at low transmission rates it will be vital to understand and control the sources of residual transmission which could, for example, be due to outdoor biting of malaria vectors when ITNs or IRS are the vector control tools relied upon for malaria elimination [36]. In this context, it is acknowledged that Sri Lanka and Iran, the two selected countries that recently achieved zero indigenous malaria cases, reported above-average conditions of malaria vector surveillance.

A strength of our paper is that it reviewed the comprehensive situation of vector control systems at national and subnational level, highlighting elements that require further attention. Our analysis indicated that, in most cases, capacity building, institutional change, and resource mobilization were key priorities that would enable a greater impact of vector control. The results can inform international agencies about the need to provide support on certain key items at subregional level, for example, through regional training courses, or to assist selected countries with their strengthening of vector control systems.

Moreover, our review provided much-needed insights into the use of the WHO guidance document for a national vector control needs assessment. The Global Vector Control Response recommended that vector-borne disease-endemic countries conduct a national vector control needs assessment and resource mobilization plan [18]. However, in 2021, only 27% of countries had completed a vector control needs assessment using the 2017 WHO guidance document [22]. We note that all assessments in our review were conducted with external financial support for national consultants and stakeholder meetings. Clearly, a limitation in conducting a (sub) national vector control needs assessment has been the time and resources required. The 2017 WHO guidance document covered topics with immediate relevance to operational programs and broader topics that are indirectly relevant to the subject matter. The latter, including policy matters, have taken a substantial part of the time and effort of the selected assessments.

To increase the feasibility and utility of national assessments, an abridged version of the guidance should be developed that is focused on operationally relevant topics. In the interim, we suggest that countries with time and resource restrictions select only those questions from the WHO guidance document that are linked to the topics of operational relevance, as

presented in Fig 1. Moreover, the methods for needs assessment should be enhanced, to make comparison between the current versus the desired situation for more precise identification of needs, and to include a process of prioritization of needs with causal analysis along the results chain; for example, by assigning weight to each need and exploring the cost of addressing the need versus the cost of inaction. The needs assessment should be complemented by guidance on resource mobilization planning. We further recommend the evaluation of whether vector control needs assessment leads to structural improvements in vector control systems over time.

Our review had several important limitations. The assessment in each country/state was conducted by a different team, which presented a source of variation in the reports. Our assignment of categories for the appraisal of perceptions was dependent on the narratives in the reports and the follow-up responses. In future, the appraisal of perceptions could potentially be simplified if the responses to the assessment's questions are provided in a standardized format, as the 'raw information' on which the narrative report is based. A limitation in the synthesis of results was the disparities in size, population and vector control conditions between countries/states. Our results probably have limited generalizability beyond South Asia and the Middle East because of differences in the disease situation, vector biology, regional strategies, and support mechanisms. This suggests that similar reviews are needed in other regions.

## Copyright statement

## Supporting information

**S1 Appendix. Panel of stakeholders involved in vector control needs assessment in each country/state.**
(DOCX)

**S2 Appendix. List of vector control needs assessment reports, by country, with availability from third party sources.**
(DOCX)

**S1 Table. Description of the key items of the vector control system.**
(DOCX)

**S1 Data. Table containing the underlying numerical data for Fig 2.**
(DOCX)

## Acknowledgments

WHO thanks Dr Subhash Lakhe, WHO Country Office, Kathmandu for facilitating the study in Nepal and gratefully acknowledges the technical inputs by stakeholders in their respective countries/states.

**Disclaimer**

The authors alone are responsible for the views expressed in this article and they do not necessarily represent the views, decisions or policies of the institutions with which they are affiliated.

## Author Contributions

**Conceptualization:** Henk van den Berg, Lauren B. Carrington, Raman Velayudhan, Rajpal S. Yadav.

**Data curation:** Henk van den Berg, Kabirul Bashar, Rajib Chowdhury, Rajendra M. Bhatt, Hardev Prasad Gupta, Ashwani Kumar, Shanmugavelu Sabesan, Ananganallur N. Shriram, Hari Kishan Raju Konuganti, Akhouri T. S. Sinha, Mohammad Mehdi Sedaghat, Ahmadali Enayati, Hameeda Mohammed Hassan, Aishath Shaheen Najmee, Sana Saleem, Surendra Uranw, Pahalagedera H. D. Kusumawathie, Devika Perera, Mohammed A. Esmail, Bhupender N. Nagpal.

**Formal analysis:** Henk van den Berg, Rajpal S. Yadav.

**Funding acquisition:** Raman Velayudhan, Rajpal S. Yadav.

**Investigation:** Henk van den Berg, Kabirul Bashar, Rajib Chowdhury, Rajendra M. Bhatt, Hardev Prasad Gupta, Ashwani Kumar, Shanmugavelu Sabesan, Ananganallur N. Shriram, Hari Kishan Raju Konuganti, Akhouri T. S. Sinha, Mohammad Mehdi Sedaghat, Ahmadali Enayati, Hameeda Mohammed Hassan, Aishath Shaheen Najmee, Sana Saleem, Surendra Uranw, Pahalagedera H. D. Kusumawathie, Devika Perera, Mohammed A. Esmail, Samira M. Al-Eryani, Roop Kumari, Bhupender N. Nagpal, Sabera Sultana.

**Methodology:** Henk van den Berg.

**Project administration:** Rajpal S. Yadav.

**Resources:** Samira M. Al-Eryani, Roop Kumari, Bhupender N. Nagpal, Sabera Sultana, Raman Velayudhan, Rajpal S. Yadav.

**Supervision:** Raman Velayudhan, Rajpal S. Yadav.

**Validation:** Henk van den Berg, Rajpal S. Yadav.

**Visualization:** Henk van den Berg.

**Writing – original draft:** Henk van den Berg.

**Writing – review & editing:** Henk van den Berg, Kabirul Bashar, Rajib Chowdhury, Rajendra M. Bhatt, Hardev Prasad Gupta, Ashwani Kumar, Shanmugavelu Sabesan, Ananganallur N. Shriram, Hari Kishan Raju Konuganti, Akhouri T. S. Sinha, Mohammad Mehdi Sedaghat, Ahmadali Enayati, Hameeda Mohammed Hassan, Aishath Shaheen Najmee, Sana Saleem, Surendra Uranw, Pahalagedera H. D. Kusumawathie, Devika Perera, Mohammed A. Esmail, Lauren B. Carrington, Samira M. Al-Eryani, Roop Kumari, Bhupender N. Nagpal, Sabera Sultana, Raman Velayudhan, Rajpal S. Yadav.

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
