## [Decision Letter · Decision Letter 0]

1 Feb 2024

Dear Dr. Yadav,

Thank you very much for submitting your manuscript "Perceived needs of disease vector control programs: A review and synthesis of (sub)national assessments from South Asia and the Middle East" for consideration at PLOS Neglected Tropical Diseases. As with all papers reviewed by the journal, your manuscript was reviewed by members of the editorial board and by several independent reviewers. The reviewers appreciated the attention to an important topic. Based on the reviews, we are likely to accept this manuscript for publication, providing that you modify the manuscript according to the review recommendations. 

Sincerely,

Philip J McCall

Academic Editor

Amy Morrison

Section Editor

Reviewer's Responses to Questions

**Key Review Criteria Required for Acceptance?**

**Methods**

-Are the objectives of the study clearly articulated with a clear testable hypothesis stated?

-Is the study design appropriate to address the stated objectives?

-Is the population clearly described and appropriate for the hypothesis being tested?

-Is the sample size sufficient to ensure adequate power to address the hypothesis being tested?

-Were correct statistical analysis used to support conclusions?

-Are there concerns about ethical or regulatory requirements being met?

Reviewer #1: (No Response)

**Results**

-Does the analysis presented match the analysis plan?

-Are the results clearly and completely presented?

-Are the figures (Tables, Images) of sufficient quality for clarity?

Reviewer #1: (No Response)

**Conclusions**

-Are the conclusions supported by the data presented?

-Are the limitations of analysis clearly described?

-Do the authors discuss how these data can be helpful to advance our understanding of the topic under study?

-Is public health relevance addressed?

Reviewer #1: (No Response)

Reviewer #2: Yes though I would like to see more input from the authors based on their experience and their knowledge of the resources needed. without it, the value of the info collected is undermined to some extent.

**Editorial and Data Presentation Modifications?**

Reviewer #1: (No Response)

Reviewer #2: N/A

**Summary and General Comments**

Reviewer #1: The manuscript explores the perceived needs of disease vector control programs in the South Asia and Middle East zones. Addressing vector control issues is crucial, particularly in regions where vector-borne diseases (VBD) are not yet under control. For those regions where control is established, more surveillance is essential to prevent resurgence. However, the presentation of the paper could be adjusted to better emphasize its contribution in transforming the pooling of country reports on the assessment of vector control needs into a research manuscript.

To enhance the manuscript's value, consider dividing it into two parts. The first part could focus on reviewing the state of the art regarding vector control in the countries covered. Important literature, including information about vector control history, current status, insecticides used, application periods, and resistance to insecticides, may provide a more comprehensive context.

The second part could present the synthesis of the national assessment per country, as it is currently structured. Countries vary in terms of vector control implementation, with only a few having programs for at least one of the three diseases. Table 1 could be more comprehensive by including the number of deaths per country and per disease. Additionally, Graph 2 needs improvement in terms of clarity and legibility. Consider using either colors or letters consistently and ensure that the legend is not redundant. Most of the terminologies and concepts used need clarification (Table 2-4).

The quantitative analysis of perception appears more qualitative, and the significant disparities between these countries raise questions about the added value of pooling data from different countries together.

While acknowledging the limitations of the study is appropriate, consider expanding on them to guide readers on parameters that should be approached with caution, particularly in the synthesis.

Reviewer #2: Viewing the scale involved, the size of the populations of the nations included, the involvement of WHO plus the magnitude/ importance and value of the NTD Information reported, the article would be unusual, but highly appropriate when published in PLOS NTDs.

PLOS authors have the option to publish the peer review history of their article (what does this mean?). If published, this will include your full peer review and any attached files.

Reviewer #1: No

Reviewer #2: No

Figure Files:

Data Requirements:

Reproducibility:

References

---

## [Editor Report · Decision Letter 1]

26 Mar 2024

Dear Dr. Yadav,

We are pleased to inform you that your manuscript 'Perceived needs of disease vector control programs: A review and synthesis of (sub)national assessments from South Asia and the Middle East' has been provisionally accepted for publication in PLOS Neglected Tropical Diseases.

Best regards,

Philip J McCall

Academic Editor

Amy Morrison

Section Editor

---

## [Editor Report · Acceptance letter]

12 Apr 2024

Dear Dr. Yadav,

We are delighted to inform you that your manuscript, "Perceived needs of disease vector control programs: A review and synthesis of (sub)national assessments from South Asia and the Middle East," has been formally accepted for publication in PLOS Neglected Tropical Diseases.

Best regards,

Shaden Kamhawi

co-Editor-in-Chief

Paul Brindley

co-Editor-in-Chief
